# Interrater Agreement and Reliability of PERCIST and Visual Assessment When Using 18F-FDG-PET/CT for Response Monitoring of Metastatic Breast Cancer

**DOI:** 10.3390/diagnostics10121001

**Published:** 2020-11-24

**Authors:** Jonas S. Sørensen, Mie H. Vilstrup, Jorun Holm, Marianne Vogsen, Jakob L. Bülow, Lasse Ljungstrøm, Poul-Erik Braad, Oke Gerke, Malene G. Hildebrandt

**Affiliations:** 1Department of Clinical Research, University of Southern Denmark, 5000 Odense, Denmark; Marianne.Vogsen@rsyd.dk (M.V.); poul-erik.braad@rsyd.dk (P.-E.B.); oke.gerke@rsyd.dk (O.G.); Malene.Grubbe.Hildebrandt@rsyd.dk (M.G.H.); 2Department of Nuclear Medicine, Odense University Hospital, 5000 Odense, Denmark; Mie.Holm.Vilstrup@rsyd.dk (M.H.V.); Jorun.holm@rsyd.dk (J.H.); jakobbulow@live.dk (J.L.B.); lasseljung@hotmail.com (L.L.); 3Department of Oncology, Odense University Hospital, 5000 Odense, Denmark; 4Odense Patient Data Explorative Network (OPEN), Odense University Hospital, 5000 Odense, Denmark; 5Centre for Personalized Response Monitoring in Oncology (PREMIO), Odense University Hospital, 5000 Odense, Denmark; 6Centre for Innovative Medical Technology (CIMT), Odense University Hospital, 5000 Odense, Denmark

**Keywords:** breast neoplasm, inter-observer, intra-observer, PERCIST, PET/CT, SULpeak, agreement

## Abstract

Response evaluation at regular intervals is indicated for treatment of metastatic breast cancer (MBC). FDG-PET/CT has the potential to monitor treatment response accurately. Our purpose was to: (a) compare the interrater agreement and reliability of the semi-quantitative PERCIST criteria to qualitative visual assessment in response evaluation of MBC and (b) investigate the intrarater agreement when comparing visual assessment of each rater to their respective PERCIST assessment. We performed a retrospective study on FDG-PET/CT in women who received treatment for MBC. Three specialists in nuclear medicine categorized response evaluation by qualitative assessment and standardized one-lesion PERCIST assessment. The scans were categorized into complete metabolic response, partial metabolic response, stable metabolic disease, and progressive metabolic disease. 37 patients with 179 scans were included. Visual assessment categorization yielded moderate agreement with an overall proportion of agreement (PoA) between raters of 0.52 (95% CI 0.44–0.66) and a Fleiss kappa estimate of 0.54 (95% CI 0.46–0.62). PERCIST response categorization yielded substantial agreement with an overall PoA of 0.65 (95% CI 0.57–0.73) and a Fleiss kappa estimate of 0.68 (95% CI 0.60–0.75). The difference in PoA between overall estimates for PERCIST and visual assessment was 0.13 (95% CI 0.06–0.21; *p* = 0.001), that of kappa was 0.14 (95% CI 0.06–0.21; *p* < 0.001). The overall intrarater PoA was 0.80 (95% CI 0.75–0.84) with substantial agreement by a Fleiss kappa of 0.74 (95% CI 0.69–0.79). Semi-quantitative PERCIST assessment achieved significantly higher level of overall agreement and reliability compared with qualitative assessment among three raters. The achieved high levels of intrarater agreement indicated no obvious conflicting elements between the two methods. PERCIST assessment may, therefore, give more consistent interpretations between raters when using FDG-PET/CT for response evaluation in MBC.

## 1. Introduction

Breast cancer (BC) is the most common cause of cancer-related deaths among women and the second most common cancer in the world [1]. Approximate 10–15% of these women will develop metastatic breast cancer (MBC) [2]. Not all patients respond alike to the same treatment and when treatment fails, which may be caused by mutation in the BC, a change in treatment will be indicated. Accurate response monitoring tools are therefore needed to direct patients to adequate treatment protocols. 

The Danish Breast Cancer Cooperative Group (DBCG) recommends the RECIST 1.1 [3] criteria for response monitoring in MBC. RECIST 1.1 relies on changes in size based on morphology of metastases as evaluated by contrast-enhanced computed tomography (CE-CT) [4]. The European Society for Medical Oncology (ESMO) recommends imaging of chest, abdomen, and bone in staging of MBC, and furthermore, imaging of target lesions every 2–4 month during treatment to evaluate response to treatment; however, they do not specify which imaging modality or response criteria should be used [5].

18F-fluorodeoxyglucose positron emission tomography/CT (FDG-PET/CT) has proven a high accuracy for diagnosing MBC and higher than conventional imaging, therefore, an increasing number of patients may be referred to departments of nuclear medicine for response evaluation with FDG-PET/CT [6]. Interpretation of FDG-PET is often made by subjective visual assessment without standardized response criteria, although the PET Response Criteria in Solid Tumors (PERCIST) were suggested by Wahl et al. [7] in 2009. The PERCIST relies on the use of 18F-FDG-PET/CT to make a semi-quantitative evaluation of the metabolic tumour response. PERCIST has been sparsely tested in clinical MBC populations, even though one study has shown that PERCIST is a superior predictor of survival in MBC when compared to RECIST 1.1 [8]. This indicates that FDG-PET/CT and PERCIST may have the potential to monitor treatment response in MBC more accurately than conventional CT and RECIST, but literature in this field is lacking. One important aspect of using FDG-PET/CT for response monitoring in MBC is information about the interrater agreement, reliability, and reproducibility in the report and classification process [9,10], but this has to our knowledge never been analysed for FDG-PET/CT with regard to assessing treatment response in patients with MBC. PERCIST proposes that the number of lesions to be measured could be either the most FDG avid lesion or five lesions as also used in RECIST 1.1 [11]. In a study by Pinker et al. [12], they found that there was no major impact on the prognostic value of response assessment when comparing the one-lesion to the five-lesions approach in patients with MBC.

This study aimed to evaluate the interrater agreement and reliability of the PERCIST and visual assessment of FDG-PET/CT in a MBC population. Furthermore, we sought to investigate the intrarater agreement when comparing visual assessment of each rater to their own respective PERCIST assessment.

## 2. Materials and Methods

This interrater agreement and reliability study is based on retrospective data from a cohort of MBC patients monitored with FDG-PET/CT at Odense University Hospital during the period 2008–2017. Study data were collected and managed using Research Electronic Data Capture (REDCap) electronic data capture tools hosted at the University of Southern Denmark [13]. REDCap is a secure, web-based application designed to support data capture for research studies. Results on the feasibility of PERCIST are being reported in a separate publication by Vogsen et al. [14]. The study has been approved by the Data Protection Agency (Journal no. 17/29850), and all subjects signed an informed consent form regarding retrospective data analysis of previously performed scans and recorded oncological files.

### 2.1. Patients

Women with MBC receiving treatment at the Department of Oncology at Odense University Hospital were eligible for the study. Identification of potential study participants took place during three months (September 2017–December 2017). The inclusion criteria were women with MBC, who were monitored by 18F-FDG-PET/CT and had at least one baseline and one follow-up scan. Patients were excluded if they were monitored by MRI, had a diagnosis of other active cancers, or if a biopsy of distant metastasis was missing.

### 2.2. FDG-PET/CT Imaging Technique

Patients were fasting for at least 4 h before receiving an intravenous injection of 4 MBq of 18F-FDG per kilogram body weight. Whole-body PET/CT scans were performed following guidelines from EANM [15] on the RX (DRX), STE (DSTE) [16], 690 (D690), 710 (D710) [17], and, MI (DMI) [18] PET/CT Discovery scanners from GE Healthcare (Waukesha, Milwaukee, WI, USA). Scans were reconstructed using iterative 3D ordered subset expectation maximization (OSEM) in matrix sizes of 128 × 128 without time-of-flight (DRX, DSTE) or 256 × 256 using time-of-flight and corrections for point-spread blurring (D690, D710 and DMI). Attenuation and scatter corrections were based on low-dose CT scans without contrast enhancement. The number of scans acquired by the DRX, DSTE, D690, D710 and DMI were 15, 33, 39, 61 and 31, respectively.

### 2.3. Visual Assessment

Three experienced specialists in nuclear medicine, all with more than ten years’ experience, made an initial qualitative visual assessment of each follow-up scan and compared it to the baseline scan. The qualitative assessment was performed without using any specific criteria, but reflected the clinical practice in our institution. If a patient changed treatment between scans, the latest scan before the change was considered as a new baseline for the subsequent treatment interval, i.e., pre-treatment-baseline. The term baseline-baseline refers in the following to the very first scan of a patient before initiating treatment for MBC. All follow-up scans were categorized into complete metabolic response (CMR), partial metabolic response (PMR), stable metabolic disease (SMD), progressive metabolic disease (PMD), mixed response (MR), or equivocal answer (EA). EA were used when e.g., all cancer lesions were stable or regressed but some new lesions had appeared which could represent cancer, unspecific activity, or benign uptake. MR were found in cases with regression in some cancer lesions, and progression or new suspect lesions in the same scan.

### 2.4. PERCIST Protocol

Due to the retrospective nature of this study, it was not possible for all scans to fully comply with the standardization procedure required of PERCIST as suggested by Wahl et al. [7]. Standardization criteria that were met in our analyses were according to matrix size and stable SUL_mean_ value in a reference volume of interest (VOI) in the liver. Hence, consecutive scans performed on scanners with alternating scan matrices were excluded as were also consecutive scans, where the SUL_mean_ in the reference VOI in the liver differed more than 20%. The following conditions could not be met in the current retrospective analysis: (a) The injection-to-scan-time should be between 50–70 min, and there must be a difference less than 15 min in the injection-to-scan-time between the baseline and the follow-up scan. (b) The same PET scanner, acquisition protocol, version of software, and reconstruction protocol must be used. (c) The difference between the injected FDG-dose must be less than 20% between the baseline and follow-up scan. (d) Patients should fast for at least four hours before injection of the FDG-dose, and their blood sugar should be less than 200 mg/dL [11]. However, we assumed that all patients fasted for more than four hours, as it is required in the local acquisition protocol, but this was not noted. Blood sugar levels were not measured routinely.

### 2.5. PERCIST Assessment

The three raters made a standardized PERCIST analysis of each follow-up scan, using the PET-VCAR software package (AW Server, version 3.2, Ext. 1.0, GE Healthcare). The software automatically found the highest SUV_max_ (of maximal 20 lesions) and assessed the cm^3^ sphere with the highest standardized uptake value corrected for lean body mass (SUL) as the SUL_peak_ value. If the automatic search missed a relevant lesion, a VOI was placed manually around the lesion, and SUL_peak_ within this VOI was automatically calculated. To this end, all relevant lesions were measured. As a tool to help the raters, automatic bookmarking of possible lesions was designated by SUV_max_. A lesion was considered measurable if it followed a pattern typical of MBC and the SUL_peak_ value within the lesion was ≥ 1.5 SUL_mean liver_ + 2 SD_liver_. In cases where no healthy liver tissue could be selected, a tubular reference VOI was placed in the aorta with a minimum threshold for evaluation of ≥ 2 SUL_mean aorta_ + 2 SD_aorta_. The SUL_peak_ was registered by all raters as well as the image ID of the single hottest/most FDG-avid metastatic lesion [7,12]. The image ID is the identification number of the axial slice of the scan, and thereby functions as a surrogate for which lesion there was measured. Moreover, it was noted if any new metastatic foci were detected, if there was unequivocal progression in a non-target lesion, or if the hottest lesion was indistinguishable from background levels as proposed by PERCIST.

The PERCIST response evaluation led to one of four response categories: (1) PMD was defined as an increase in SUL_peak_ by >30% and >0.8 SUL units, detection of new metastatic foci, or unequivocal progression in a non-target lesion. (2) PMR was defined as a decrease in SUL_peak_ by >30% and >0.8 SUL units without detection of new lesions. (3) CMR was defined as total resolution of FDG avidity and the hottest lesion was indistinguishable from background FDG levels. (4) SMD was defined as change <30% in SUL_peak_ or <0.8 SUL units without detection of new lesions [11,19]. For all raters and for both the visual assessment and PERCIST categorization, the data were dichotomized into response (CMR, PMR, or SMD) or non-response (PMD, MR, or EA). All raters assessed the scans individually while blinded to assessments of the other raters.

### 2.6. Statistical Analyses

For the agreement analysis with categorical data, proportions of agreement and Cohen’s kappa statistics were used [9]. Agreement between all three raters was measured by Fleiss’ kappa. As proposed by Landis and Koch [20], we considered the strength of agreement by kappa values <0.00 as ‘poor’, 0.00–0.20 as ‘slight’, 0.21–0.40 as ‘fair’, 0.41–0.60 as ‘moderate’, 0.61–0.80 as ‘substantial’, and 0.81–1.00 as ‘almost perfect’. Ninety-five percent confidence intervals (95% CIs) for proportions of agreement were based on the Wilson-score method, except for the overall intrarater agreement between visual assessment and PERCIST assessment of each rater; here, Wald-type 95% CIs were derived from simple regression analysis using clustered sandwich estimators in order to account for the correlation structure in the data (raters applied both visual assessment and PERCIST to the same scans). Ninety-five percent confidence intervals for kappa values were derived with bootstrapping techniques. For differences in agreement, *p* < 0.05 was considered significant. To compare and illustrate the differences in the measured SUL_peak_ values and image IDs, we generated Bland-Altman (BA) plots [9,21,22,23] for both baseline and post-baseline. For post-baseline data, we excluded scans with PERCIST categorization of CMR, as it by definition has no distinctive lesion to measure. Statistical analyses were performed using STATA/IC 15.0 (StataCorp, College Station, TX, USA).

## 3. Results

Two-hundred-and-twenty-eight patients were identified for potential study inclusion from which 119 women did not meet the inclusion criteria, and further 66 women were excluded for various reasons as seen in Figure 1. Of 43 patients with 315 scans, 136 scans were considered non-comparable as they did not meet our standardization criteria. This left a sample of 179 scans (56.8%) in 37 patients with a median age of 65 years (range 33.5–86.5) and a median of 4.8 (range 0.3–25.6) years until relapse since primary BC diagnosis. The characteristics and the receptor status (Estrogen receptor and Human Epidermal Growth Receptor 2) of the primary tumors and the verifying biopsy from metastatic lesions are shown in Appendix A. The systemic therapies of the patients often changed (due to progression of disease or side effects) between endocrine therapy, chemotherapy, Anti-HER2, and sometimes in combination. It also changed which medications where used in each type of therapy.

### 3.1. Visual Assessment

There was moderate agreement between raters for visual assessment with a Fleiss kappa of 0.54 (95% CI 0.46–0.62) (Table 1), and the overall proportion of agreement was 0.52 (95% CI 0.44–0.66). Response classification for visual assessment has been summarized in Table 2 for pairs of raters. Data were missing for 4 and 3 assessments for rater 1 and rater 2, respectively, two of these were from the same scan.

### 3.2. Differences in Baseline SUL_peak_ Values and Image IDs

Interrater differences were identified at baseline (either baseline-baseline or pre-treatment- baseline) in six scans of six patients. Of these, in two scans all raters measured the same SUL_peak_ value in using different image IDs. On two occasions all raters used the same image ID, but one rater measured a different SUL_peak_ compared to the other raters. In two cases, each rater used different image IDs, but two raters measured the same SUL_peak_. Differences in image IDs were identified with BA plots which, in turn, cannot be interpretated otherwise (Appendix A).

The BA plots for the differences in SUL_peak_ of the 62 baseline scans are illustrated in Figure 2. Rater 1 and rater 2 had the widest 95% limits of agreement (LoA) (−0.68 to 0.69), whereas rater 2 and rater 3 had the narrowest (BA LoA: −0.15 to 0.13).

### 3.3. Differences in Post-Baseline SUL_peak_ Values and Image IDs

Interrater differences were identified in 18 (11 patients) of 123 post-baseline scans. There were six instances, where each rater measured the same SUL_peak_ value in different image IDs, and four occasions where all raters used the same image ID, but one rater measured a different SUL_peak_ compared to that of the others. In four cases, one rater differed in SUL_peak_ value and image ID, whereas the two others agreed. There were four cases, where all raters disagreed in both SUL_peak_ and image IDs (Appendix A). Two patients contributed with eight of the 18 scans. For SUL_peak_, rater 1 and rater 2 had the widest BA LoA of −0.59 to 0.64), and smallest for rater 2 and rater 3 (BA LoA: −0.42 to 0.37) (Figure 3).

### 3.4. PERCIST Assessment

There was substantial agreement between raters for PERCIST assessment with a Fleiss kappa estimate of 0.68 (95% CI 0.60–0.75) and the overall proportion of agreement for the PERCIST group was 0.65 (95% CI 0.57–0.73) (Table 1). Response classification using PERCIST is summarized in Table 3 for pairs of raters and shows respective discordant results.

In six scans of two patients the SUL_peak_ value alone differed enough from the baseline value to make a change in PERCIST categorization, in two of the scans in one patient the rating changed between SMD and PMD, and in four scans of the other patient the rating changed between PMR and SMD, this was due to the raters had chosen different lesions at baseline. Eight cases of disagreement between CMR and PMR, and CMR and SDM categorization were identified because the raters disagreed about the total resolution of FDG avidity. In seven other cases, different interpretation of the detection of new metastatic foci was the reason for the difference in PERCIST categorization. In eight other cases, different evaluations of unequivocal progression in a non-target lesion caused a difference in PERCIST categorization. In 20 additional cases the raters differed in both the detection of new metastatic foci and detection of unequivocal progression in a non-target lesion.

### 3.5. Response versus Non-Response

Discordant pairs of assessments between visual assessment and PERCIST are shown by rater in Appendix A. Using visual response/non-response assessment, the overall proportion of agreement was 0.68 (95% CI 0.60–0.75) and moderate agreement was suggested by a Fleiss kappa of 0.51 (95% CI 0.39–0.63; Table 1). With PERCIST, the overall proportion of agreement of was 0.74 (95% CI 0.66–0.80) and substantial agreement was indicated according to a Fleiss kappa of 0.61 (95% CI 0.50–0.72; Table 1).

### 3.6. Differences in Agreement and Reliability Measures

The difference in proportions of agreement between overall estimates for PERCIST and visual assessment was 0.13 (95% CI 0.06–0.21; *p* = 0.001) in favor of PERCIST, that of kappa was 0.14 (95% CI 0.06–0.21; *p* < 0.001). For the response/non-response categorization, the difference in proportions of agreement was 0.06 (95% CI 0.004–0.13; *p* = 0.04) in favor of PERCIST, that of kappa was 0.11 (95% CI 0.02–0.21; *p* = 0.02). A summary of agreement and reliability measures is shown in Table 1.

### 3.7. Intrarater Agreement

The intrarater agreement when comparing visual assessment of each rater to their own respective PERCIST assessment (Appendix A) showed the overall proportion of agreement to be 0.80 (95% CI 0.75–0.84) and a substantial agreement with a Fleiss kappa of 0.74 (95% CI 0.69–0.79). The overall intrarater comparison between PERCIST and visual assessment in respect of response/non-response resulted in a proportion of agreement of 0.92 (95% CI 0.89–0.95) and achieved almost perfect agreement with a Fleiss kappa of 0.82 (95% CI 0.77–0.88).

## 4. Discussion

In this retrospective clinical study of patients with metastatic breast cancer, we found that raters achieved a significantly higher proportion of agreement and reliability when using PERCIST evaluation compared to visual assessment of FDG-PET/CT for response monitoring. This indicated that a rater was more likely to assign the same response category to a scan as the other raters when using PERCIST compared with using visual assessment.

To provide e.g., oncologists a useful and clinically more relevant evaluation/response category, patients were dichotomized into responders/non-responders. Raters agreed significantly more often when using PERCIST than when using visual assessment. However, in nearly every fourth patient, one rater disagreed with the others in whether a patient was a responder or nonresponder, when using PERCIST. This kind of disagreement could have a major effect on later patient management and the decision to change treatment. In several cases the reason for disagreement was on account of the qualitative aspect of PERCIST, i.e., the disagreement was based upon whether a rater believed to have detected new metastatic foci, unequivocal progression in a non-target lesion, or total resolution of FDG avidity and the hottest lesion was indistinguishable from background FDG levels. These qualitative components overrule the quantitative SULpeak measurements in regards to the final PERCIST categorization. When looking at the discordance between raters in measured SUL_peak_ and selected image IDs, we found that raters differed in several scans regarding which metastatic lesion they selected for SUL_peak_ measurement, resulting in different SUL_peak_ values. One example is shown in Figure 4; there are multiple avid masses shown in the maximum intensity projection (MIP) in lung, liver, and thyroid. All raters interpreted the lesion in the thyroid as a new primary disease and not a metastasis. Rater 1 and rater 3 interpreted the lung lesions as metastatic lesions, however, rater 2 interpreted these as pneumonia and measured SUL_peak_ for the liver lesion instead. A consequence of this could be if one rater measured a different SUL_peak_ value on a baseline scan than the other raters but measured identically on follow-up scans. Thus, it is possible that the PERCIST categorization would differ accordingly. This is because the PERCIST categorization depends on changes in SUL_peak_ values related to the baseline scan. High levels of the intrarater agreement were found, when PERCIST for each rater was matched to the visual assessment for the same rater indicating no obvious conflicting elements between the two methods.

A weakness of this study was that we included scans even though they did not fully fulfill the standardization procedure required by PERCIST [7], and, therefore, we cannot be completely sure the data is reliable and reproducible. A relatively large proportion of scans were excluded due to non-comparability, and this may give a selection bias. This exclusion was because that we believe that the risk of variations in SUL_peak_ were due to other circumstances than actual variation in the cancer lesions i.e., scans performed on scanners with different matrix sizes (128 × 128 vs 256 × 256), and scans where the SUL_mean_ in the liver differed more than 20%.

Another potential selection bias may be associated with the reason for a patient to be monitored by 18F-FDG-PET/CT as compared to CE-CT, which is the preferred alternative method for response monitoring in MBC at our institution. Reasons for choosing 18F-FDG-PET/CT could be a substantial tumour burden, FDG-positive metastatic lesions, or a matter of preference of the individual oncologist. A further potential limitation of the study may be that the raters were aware that their evaluations would be compared to other raters, and therefore this might have influenced their answers i.e., the Hawthorne effect [24]. Our study uses longitudinal response monitoring and evaluation in a retrospective clinical population, which is considered a strength as it mimics daily clinical practice.

We did not use any specific criteria for the visual assessment which is not routinely done in our institution, and this could explain the lower interobserver agreement for visual assessment and may therefore represent the major limitation of our study. However, the detection of new FDG-avid lesions suspicious of cancer will typically be interpreted as progressive disease as well as the subjectively interpreted progression of a previously detected lesion. Although we do not use quantitative measures in our routine practice, this is not prohibited and could be applied; but still this was not allowed for visual assessment in the present analysis. It is our impression that qualitative interpretations are much more widely used in clinical practice than (semi)-quantitative methods, and therefore we think that the qualitative assessment in the present analysis would reflect current clinical practice in most institutions.

No other study has, to the best of our knowledge, investigated the interrater agreement or reliability of the PERCIST in MBC. However, one study by Fledelius et al. [10] found that in non-small cell lung cancer, eight observers achieved ‘substantial agreement’ for PERCIST and ‘moderate agreement’ for qualitative visual assessment. This correlates well with our results, although, they did not have a longitudinal study design. Other studies have shown for various types of cancer that raters generally achieved ‘moderate agreement’ for qualitative visual assessment [25,26,27], and ‘almost perfect agreement’ for quantitative assessment when measuring the standardized uptake value (SUV) [28,29,30,31,32]. This supports that SUV (from which SUL is derived) is a robust variable to measure; therefore, the discordance in interrater agreement for PERCIST is likely due to the qualitative judgment (detection of new foci or unequivocal progression) of the raters.

This study suggests that national and international guidelines should consider recommendations on standardization tools for acquisition and analysis of 18F-FDG-PET/CT with the purpose of response monitoring [33]. Standardized response assessment seems to yield higher agreement among raters, and it allows, therefore, a more distinct assessment of treatment effects. In the absence of a reliable reference method of the true response categorization, relatively high reliability and reproducibility of PERCIST encourages its use.

The PERCIST response categorization proposes a comparison of a follow-up scan to the baseline scan, only. There can be cases where follow-up scans would be categorized as PMR when compared to baseline, but if it was compared to a previous scan with a decreased SUL_peak_ of the target lesion, it would be categorized as PMD [34]. Therefore, a response evaluation categorization using comparison to baseline scans only may be misleading, and we believe that comparison to a nadir scan would give a more correct and nuanced picture of the disease fluctuation in longitudinal response monitoring. With quantification it will be possible to identify the nadir scan which is important in cases where patients with regressive diseases eventually experience progressive disease. Adding PERCIST may change clinical decision making, but it was not possible to analyze in a retrospective study design, and we suggest a future prospective study that investigate the impact of PERCIST in clinical decision making.

In conclusion, the semi-quantitative PERCIST assessment achieved a significantly higher overall level of agreement and reliability when compared with qualitative visual assessment of 18F-FDG-PET/CT for response evaluation in patients with metastatic breast cancer. Hence, this retrospective study simulating everyday clinical practice indicated PERCIST to be a robust and feasible method yielding more consistent interpretations between raters when compared with qualitative visual assessment. To this end, appropriately sized prospective multicentre studies are warranted for evaluation of the clinical impact of using standardized assessment in this patient setting.

## Figures and Tables

**Figure 1 diagnostics-10-01001-f001:**
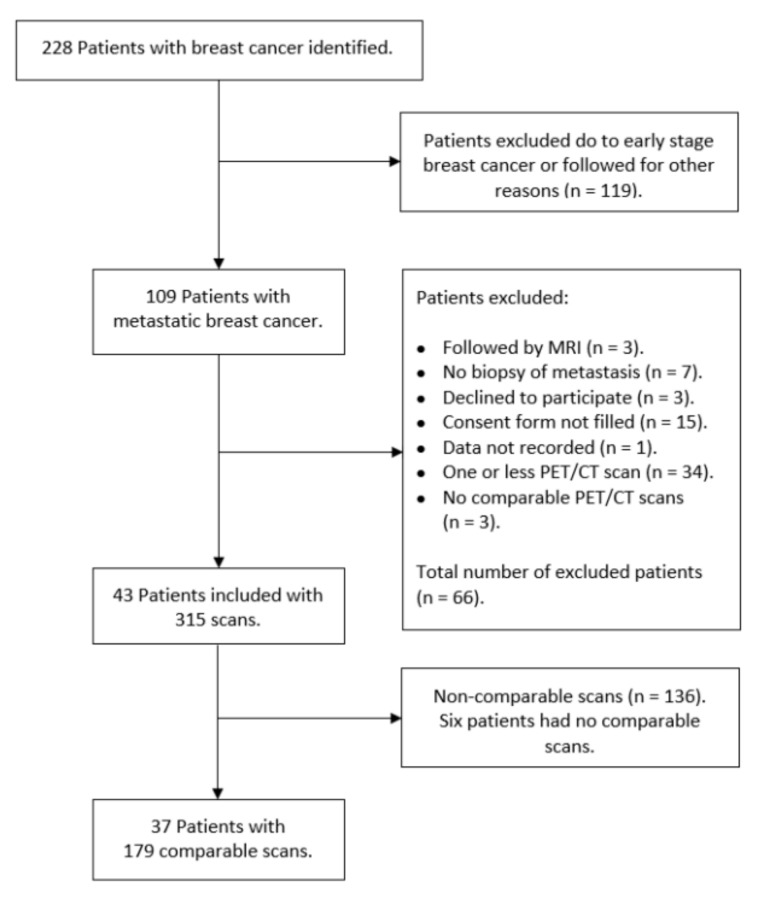
Flowchart over included patients and 18F-FDG-PET/CT scans.

**Figure 2 diagnostics-10-01001-f002:**
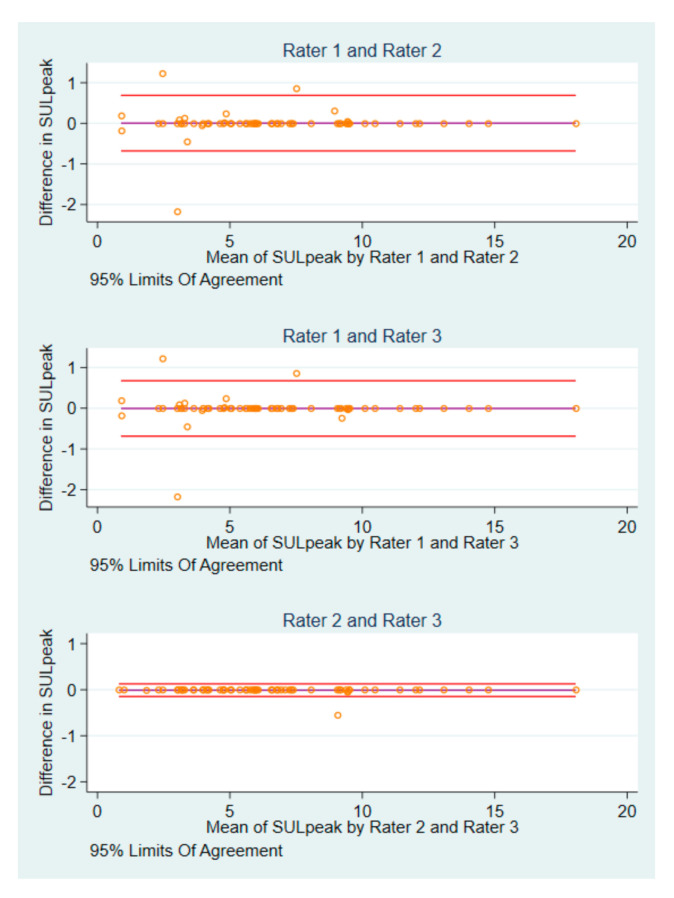
BA plots of SUL_peak_ baseline measurements by pairs of raters.

**Figure 3 diagnostics-10-01001-f003:**
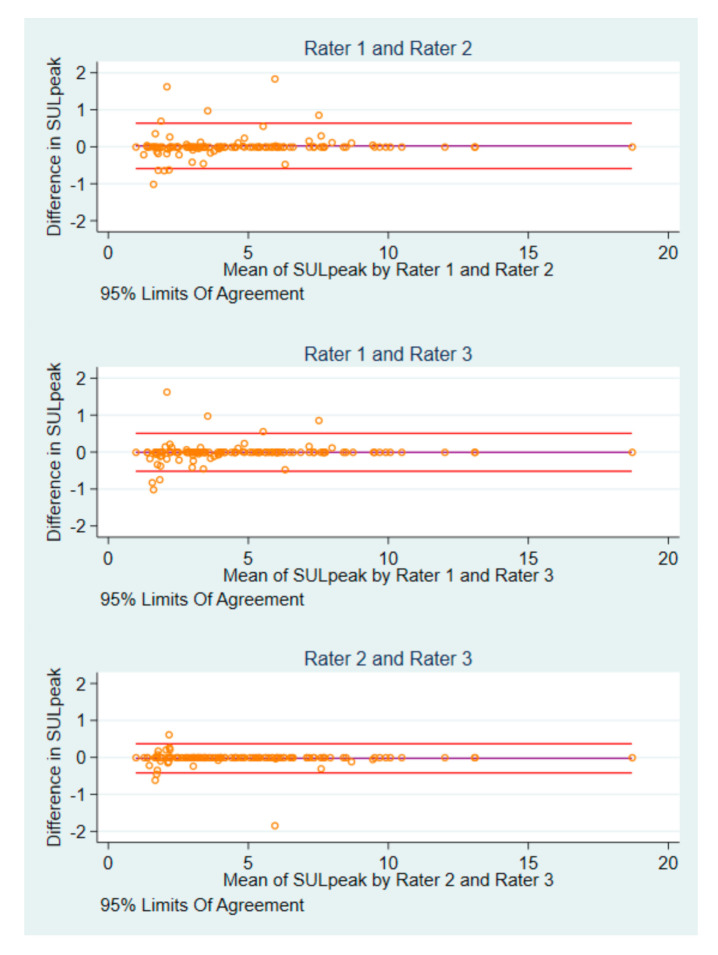
Bland-Altman plots of post-baseline SUL_peak_ measurements by pairs of raters.

**Figure 4 diagnostics-10-01001-f004:**
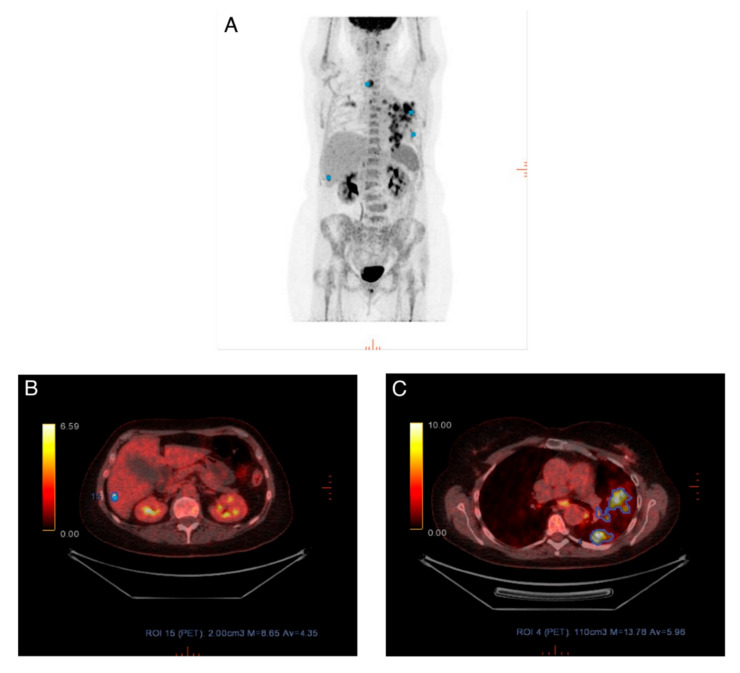
(**A**) Maximum intensity projection (MIP) for scan where different lesions were measured. (**B**) Rater 2 measured SUL_peak_ in a lesion in the liver, as she interpreted the lung lesion as a non-metastatic lesion. (**C**) Rater 1 and rater 3 interpreted the lung lesion as metastatic and measured SUL_peak_ in this.

**Table 1 diagnostics-10-01001-t001:** Summary of agreement and reliability measures.

Criterion	Comparison of Raters	Proportion of Agreement		Kappa	
		Point Estimate	95% CI	Point Estimate	95% CI
PERCIST	1–2	0.75	0.67–0.82	0.65	0.55–0.75
	1–3	0.77	0.68–0.83	0.69	0.59–0.78
	2–3	0.78	0.70–0.84	0.70	0.60–0.79
	All	0.65	0.57–0.73	0.68	0.60–0.75
Visual assessment	1–2	0.68	0.60–0.75	0.58	0.47–0.69
	1–3	0.65	0.57–0.73	0.54	0.43–0.65
	2–3	0.64	0.56–0.71	0.52	0.42–0.62
	All	0.52	0.44–0.60	0.54	0.46–0.62
PERCIST vs. Visual assessment	1–1	0.79	0.71–0.85	0.73	0.64–0.82
	2–2	0.79	0.72–0.85	0.74	0.65–0.82
	3–3	0.82	0.74–0.87	0.74	0.65–0.83
	All	0.80	0.75–0.84	0.74	0.69–0.79
Response vs. Non-response: PERCIST	1–2	0.79	0.71–0.85	0.55	0.41–0.68
	1–3	0.85	0.78–0.90	0.66	0.52–0.73
	2–3	0.84	0.77–0.89	0.64	0.51–0.77
	All	0.74	0.66–0.80	0.61	0.50–0.72
Response vs. Non-response: Visual assessment	1–2	0.79	0.72–0.85	0.56	0.41–0.70
	1–3	0.81	0.74–0.87	0.53	0.37–0.68
	2–3	0.76	0.68–0.82	0.47	0.33–0.62
	All	0.68	0.60–0.75	0.51	0.39–0.63
Response vs. Non-response: PERCIST vs. Visual	1–1	0.94	0.89–0.97	0.87	0.78–0.96
	2–2	0.92	0.86–0.95	0.84	0.74–0.93
	3–3	0.91	0.86–0.95	0.77	0.65–0.89
	All	0.92	0.89–0.95	0.82	0.77–0.88

95% CI: 95% confidence interval.

**Table 2 diagnostics-10-01001-t002:** Interrater response assessment using visual assessment.

**Rater 1**	**Rater 2**						**Total (%)**
	**CMR (%)**	**PMR (%)**	**SMD (%)**	**PMD (%)**	**MR (%)**	**EA (%)**	
CMR (%)	**16 (11.85)**	1 (0.74)	0 (0)	0 (0)	0 (0)	0 (0)	17 (12.59)
PMR (%)	1 (0.74)	**37 (27.41)**	1 (0.74)	9 (6.67)	0 (0)	2 (1.48)	50 (37.04)
SMD (%)	0 (0)	7 (5.19)	**8 (5.93)**	7 (5.19)	0 (0)	1 (0.74)	23 (17.04)
PMD (%)	0 (0)	7 (5.19)	2 (1.48)	**34 (25.19)**	2 (1.48)	0 (0)	45 (33.33)
MR (%)	0 (0)	0 (0)	0 (0)	0 (0)	**0 (0)**	0 (0)	0 (0)
EA (%)	0 (0)	0 (0)	0 (0)	0 (0)	0 (0)	**0 (0)**	0 (0)
Total (%)	17 (12.59)	52 (38.52)	11 (8.15)	50 (37.04)	2 (1.48)	3 (2.22)	135 (100)
**Rater 1**	**Rater 3**						**Total (%)**
	**CMR (%)**	**PMR (%)**	**SMD (%)**	**PMD (%)**	**MR (%)**	**EA (%)**	
CMR (%)	**12 (8.76)**	4 (2.92)	0 (0)	0 (0)	0 (0)	1 (0.73)	17 (12.41)
PMR (%)	2 (1.46)	**40 (29.20)**	6 (4.38)	2 (1.46)	0 (0)	0 (0)	50 (36.50)
SMD (%)	0 (0)	7 (5.11)	**16 (11.68)**	1 (0.73)	0 (0)	0 (0)	24 (17.52)
PMD (%)	0 (0)	9 (6.57)	13 (9.49)	**24 (17.52)**	0 (0)	0 (0)	46 (33.58)
MR (%)	0 (0)	0 (0)	0 (0)	0 (0)	**0 (0)**	0 (0)	0 (0)
EA (%)	0 (0)	0 (0)	0 (0)	0 (0)	0 (0)	**0 (0)**	0 (0)
Total (%)	14 (10.22)	60 (43.80)	35 (25.55)	27 (19.71)	0 (0)	1 (0.73)	137 (100)
**Rater 2**	**Rater 3**						**Total (%)**
	**CMR (%)**	**PMR (%)**	**SMD (%)**	**PMD (%)**	**MR (%)**	**EA (%)**	
CMR (%)	**12 (8.7)**	4 (2.90)	0 (0)	0 (0)	0 (0)	1 (0.72)	17 (12.32)
PMR (%)	1 (0.72)	**43 (31.16)**	8 (5.80)	0 (0)	0 (0)	0 (0)	52 (37.68)
SMD (%)	0 (0)	2 (1.45)	**8 (5.80)**	1 (0.72)	0 (0)	0 (0)	11 (7.97)
PMD (%)	1 (0.72)	9 (6.52)	16 (11.59)	**27 (19.57)**	0 (0)	0 (0)	53 (38.41)
MR (%)	0 (0)	0 (0)	2 (1.46)	0 (0)	**0 (0)**	0 (0)	2 (1.45)
EA (%)	0 (0)	2 (1.45)	1 (0.72)	0 (0)	0 (0)	**0 (0)**	3 (2.17)
Total (%)	14 (10.14)	60 (43.48)	35 (25.36)	28 (20.29)	0 (0)	1 (0.72)	138 (100)

The Data in bold shows exact agreement among the pairs of raters. CMR: Complete metabolic response; EA: Equivocal answer; MR: Mixed response; PMD: Progressive metabolic disease; PMR: Partial metabolic response; SMD: Stable metabolic response.

**Table 3 diagnostics-10-01001-t003:** Interrater response assessment using PERCIST.

**Rater 1**	**Rater 2**				**Total (%)**
	**CMR (%)**	**PMR (%)**	**SMD (%)**	**PMD (%)**	
CMR (%)	**17 (12.06)**	1 (0.71)	0 (0)	0 (0)	18 (12.77)
PMR (%)	0 (0)	**34 (24.11)**	4 (2.84)	11 (7.80)	49 (34.75)
SMD (%)	0 (0)	0 (0)	**17 (12.06)**	7 (7.96)	24 (17.02)
PMD (%)	0 (0)	3 (2.13)	9 (6.38)	**38 (26.95)**	50 (35.46)
Total (%)	17 (12.06)	38 (26.95)	30 (21.28)	56 (39.72)	141 (100)
**Rater 1**	**Rater 3**				**Total (%)**
	**CMR (%)**	**PMR (%)**	**SMD (%)**	**PMD (%)**	
CMR (%)	**13 (9.22)**	4 (2.84)	1 (0.71)	0 (0)	18 (12.77)
PMR (%)	2 (1.42)	**40 (28.37)**	4 (2.84)	3 (2.13)	49 (34.75)
SMD (%)	0 (0)	0 (0)	**22 (15.60)**	2 (1.42)	24 (17.02)
PMD (%)	0 (0)	4 (2.84)	12 (8.51)	**34 (24.11)**	50 (35.46)
Total (%)	15 (10.64)	48 (34.04)	39 (27.66)	39 (27.66)	141 (100)
**Rater 2**	**Rater 3**				**Total (%)**
	**CMR (%)**	**PMR (%)**	**SMD (%)**	**PMD (%)**	
CMR (%)	**12 (8.51)**	4 (2.84)	1 (0.71)	0 (0)	17 (12.06)
PMR (%)	3 (2.13)	**34 (24.11)**	0 (0)	1 (0.71)	38 (26.95)
SMD (%)	0 (0)	0 (0)	**28 (19.86)**	2 (1.42)	30 (21.28)
PMD (%)	0 (0)	10 (7.09)	10 (7.09)	**36 (25.53)**	56 (39.72)
Total (%)	15 (10.54)	48 (34.04)	39 (27.66)	39 (27.66)	141 (100)

The Data in bold shows exact agreement among the pairs of raters. CMR: Complete metabolic response; PMD: Progressive metabolic disease; PMR: Partial metabolic response; SMD: Stable metabolic response.

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
