# Peer review of "Interrater Agreement and Reliability of PERCIST and Visual Assessment When Using 18F-FDG-PET/CT for Response Monitoring of Metastatic Breast Cancer"

_diagnostics, 2020, doi:10.3390/diagnostics10121001_

Round 1

Reviewer 1 Report

Authors compared the interrater agreement between visual assessment and semiquantitative assessment using RECIST. 

The higher agreement found with semiquantitative assessment is mainly based on the use of  defined criteria. The visual assessment in the response evaluation is not reliable if no previous interpretation criteria are defined. So, to evaluate PET  without using any specific criteria is the main limitation and explain the lower interobserver agreement. 

Author Response

We thank the reviewer for pointing this out, we have integrated this aspect in the added paragraph in the Discussion (lines 312-321).

Reviewer 2 Report

The manuscript discusses the inter- and intra reader variation in classifying metabolic response of metastatic breast cancer to systemic therapy. This is clinically relevant, as the use of FDG PET for this purpose is increasing, while such evaluations of interpretation have not been performed to my knowledge.

The manuscript is well-written, the evaluations are straightforward, and the conclusions are supported by the presented evidence. The limitations of the study and correlation with prior evidence are adequately described.

I have only one comment, that may help to further improve the manuscript:

- There is no indication of the types of breast cancer that were included (e.g. her2 positive etc) and the applied systemic therapies (various chemos, receptor targeting, hormonal, or perhaps even immune therapies). The biological responses of tumour and normal tissues to these treatments may vary significantly, with implications for metabolic response as well as the occurrence of false-positive findings (e.g. inflammation). I think it would help interpretation of the data when a table of the 37 treated patients and their treatments is added.

Author Response

We thank the reviewer for this comment. The characteristics and receptor status of the primary tumours and the verifying biopsy from metastatic lesions are shown in the supplementary Table S1. The reference to Table S1 has been more clearly made in the Results (lines 183-185). The treatment of each of the 37 patients are difficult to display as in this longitudinal study the treatment often changed (due to progression of disease or side effects) between endocrine therapy, chemotherapy, Anti-HER2, and sometimes in combination. It also changed which (and multiple) medications where used in each type of therapy. This is now mentioned in the Results (lines 185-188). We argue that there is no difference in the way the evaluation is done in relation to the different systemic treatment used, despite the biological responses of tumour and normal tissues to these treatments may vary.

Reviewer 3 Report

This study evaluates an important question about the interrater agreement and reliability of visual assessment and PERCIST in metastatic breast cancer patients. FDG PET interpretation is often criticized for high variability between readers and this perception may be one of the reasons why PERCIST has not been widely adopted for clinical trials - due to concerns of inconsistent response assessments.   

I think this study is well-designed, executed, and scientifically and statistically sound. Manuscript is overall well-written.

Just a few comments/suggestions:

Methods: "The qualitative assessment was performed without using any specific criteria, but reflected the clinical practice in our institution." I think this statement needs to be clarified further.

Did the readers just look at the PET scans and make a judgement subjectively about the response classification? Were they allowed to make any measurements (not in the manner of PERCIST), but for example, measure the SUVmax of few lesions to guide their assessment or not measurements were allowed? Did any of the readers have their own internal criteria for visual assessment, i.e. lesions hotter than liver background?

The interrater response assessment tables 2 and 3 and breakdown of the agreement among different response categories is interesting and I suggest you expand upon it on the Results and Discussion section, perhaps highlighting any patterns. For example, one would expect considerable overlap in PMR and SMD, however there are a fair amount of cases in which one reader classified as PMR and another classified as PMD, which are near opposite sides of spectrum (i.e. Reader 1 vs. Reader 2, 11 scans -7.8%) and have significant impact on patient managment? What accounts for this, if there is an explanation?

Author Response

This study evaluates an important question about the interrater agreement and reliability of visual assessment and PERCIST in metastatic breastcancer patients. FDG PET interpretation is often criticized for high variability between readers and this perception may be one of the reasons why PERCIST has not been widely adopted for clinical trials -due to concerns of inconsistent response assessments.

I think this study is well-designed, executed, and scientifically and statistically sound. Manuscript is overall well-written.

Just a few comments/suggestions:

Methods: "The qualitative assessment was performed without using any specific criteria, but reflected the clinical practice in our institution." I think this statement needs to be clarified further.

Did the readers just look at the PET scans and make a judgement subjectively about the response classification? Were they allowed to make any measurements (not in the manner of PERCIST), but for example, measure the SUVmax of few lesions to guide their assessment or not measurements were allowed? Did any of the readers have their own internal criteria for visual assessment, i.e. lesions hotter than liver background?

Reply: We thank the reviewer for the overall impression of the manuscript and for pointing out the need for clarification about the visual assessment as also pointed out by the editor and Reviewer #1. We have integrated this aspect in the added paragraph in the Discussion (lines 312-321).

The interrater response assessment tables 2 and 3 and breakdown of the agreement among different response categories is interesting and I suggest you expand upon it on the Results and Discussion section, perhaps highlighting any patterns. For example, one would expect considerable overlap in PMR and SMD, however there are a fair amount of cases in which one reader classified as PMR and another classified as PMD, which are near opposite sides of spectrum (i.e. Reader 1 vs. Reader 2, 11 scans -7.8%) and have significant impact on patient management? What accounts for this, if there is an explanation?

Reply: We once more thanks the reviewer for this insightful comment. Unfortunately,the reasons behind each visual assessment shown in table 2 is not available. However, a breakdown of table 3 and the reasons for different categorization has been added in the Results (lines 223-233)and an explanation of the disagreement and the impact of patient management have been added in the Discussion (lines 269-278).

Round 2

Reviewer 1 Report

The same that in the previous review